# Biotechnological and Technical Challenges Related to Cultured Meat Production

**Davide Lanzoni** [1,†], **Filippo Bracco** [2,†], **Federica Cheli** [1,3], **Bianca Maria Colosimo** [2], **Davide Moscatelli** [4], **Antonella Baldi** [1], **Raffaella Rebucci** [1,*] **and Carlotta Giromini** [1,3]

1 Department of Veterinary and Animal Sciences (DIVAS), Università degli Studi di Milano, Via dell'Università 6, 29600 Lodi, Italy; davide.lanzoni@unimi.it (D.L.); federica.cheli@unimi.it (F.C.); antonella.baldi@unimi.it (A.B.); carlotta.giromini@unimi.it (C.G.)
2 Department of Mechanical Engineering, Politecnico di Milano, Via La Masa 1, 20156 Milano, Italy; filippo.bracco@polimi.it (F.B.); biancamaria.colosimo@polimi.it (B.M.C.)
3 CRC "Innovation for Well-Being and Environment (I-WE)", Università degli Studi di Milano, Via Festa del Perdono 7, 20122 Milano, Italy
4 Department of Chemistry, Materials and Chemical Engineering "Giulio Natta", Piazza Leonardo da Vinci 32, 20133 Milano, Italy; davide.moscatelli@polimi.it
* Correspondence: raffaella.rebucci@unimi.it
† These authors contributed equally to this work.

**Abstract:** The constant growth of the population has pushed researchers to find novel protein sources. A possible solution to this problem has been found in cellular agriculture, specifically in the production of cultured meat. In the following review, the key steps for the production of in vitro meat are identified, as well as the most important challenges. The main biological and technical approaches are taken into account and discussed, such as the choice of animal, animal-free alternatives to fetal bovine serum (FBS), cell biomaterial interactions, and the implementation of scalable and sustainable biofabrication and culturing systems. In the light of the findings, as promising as cultured meat production is, most of the discussed challenges are in an initial stage. Hence, research must overcome these challenges to ensure efficient large-scale production.

**Keywords:** cultivated meat; cellular agriculture; FBS alternatives; 3D scaffolding; 3D bioprinting; edible bioink; sustainability; in vitro meat

## 1. Introduction

The current global population is approximately 8 billion, a number that is set to grow rapidly by 2050, when it is estimated that the number of people on Earth will reach 9–11 billion [1]. Such a significant population increase will lead to an exponential growth in the demand for food products, reaching 50% by 2030 and, doubling by 2050, at which point the demand will be difficult to meet without negatively impacting environmental health [2,3]. Products subject to high demand are those of animal origin, especially meat and dairy products. In 2012, the Food and Agriculture Organization (FAO) estimated that the global demand for meat is expected to reach 455 million tons by 2050, a 76% increase from 2005. This trend also applies to fish farming, where the amount of fish obtained from aquaculture has increased from 4.7 to 66.6 million tons in just 32 years, and the global demand is expected to reach 140 million by 2054 [2,4].

Consequently, to meet the massive demand for animal products, intensive farming has been largely applied. Despite their success after the industrialization process of the 1960s, animal production systems have proven to be extremely fragile.

Current industrial-scale breeding systems have been the subject of debate regarding animal welfare, environmental protection, and public health.

Since the 1980s, intensive livestock farming has been the source of numerous public health crises, such as mad cow disease and, later, the contamination of farmed chicken, beef,

pork, milk, and salmon products with dioxin, and more recently, swine flu and poultry flu [2]. Another problem related to intensive livestock farming is the environmental impact. The intensification of livestock and agricultural production is involved in greenhouse gas (GHG) emissions, particularly methane ($CH_4$) and nitrous oxide ($N_2O$). As reported by Guerci et al., agriculture contributes 10–12% of all total emissions, while livestock production, especially ruminants, contribute a total of 14,5% [5,6]. However, this statement is at odds with what Chriki and Hocquette reported. According to these authors, while it is true that livestock has an environmental impact, especially in terms of $CH_4$, $N_2O$, and carbon dioxide ($CO_2$) emissions, it is also true that the latter is the main source of emissions of in vitro meat due to the fossil energy consumed to ensure cell growth and proliferation [7]. In support of this contention, Lynch et al. confirmed how the impact of cultured meat will be minor only in the short term (within 20 years), but not in the long term (beyond 100 years), since $CO_2$, unlike $CH_4$, accumulates for long periods within the atmosphere [8].

Moreover, although livestock, mostly ruminants, consume food not intended for humans, they occupy about 70% of global agricultural land and consume about 35% of agricultural crops, directly competing with the production of crops for human consumption and with potential alternative land uses, including nature conservation [9]. The agricultural sector is extremely resource-rich and continues to transform with population growth. Global food production is the largest user of freshwater and uses about 38% of the land. The remaining 62% of the global land area is estimated to be unsuitable for cultivation due to climate, topography, poor soil quality, urban development, or because it is covered by natural lands such as forests. Therefore, little arable land remains for agricultural expansion without a negative impact on environmental health [10].

*Cellular Agriculture and Cultured Meat*

Industrial biotechnology holds a possible key to providing humanity with nutritious, safe, and healthy food while minimizing the use of resources such as energy, water, and land [3]. This solution is called "cellular agriculture" and involves producing food products such as meat or fish from single cells rather than whole organisms such as animals, whose priority is to manufacture products that are similar at the molecular level to those made by traditional techniques [11]. With cellular agriculture methods it is possible to produce artificial meat, also called cultured or clean meat, through the differentiation of muscle satellite cells in vitro [12].

Cultured meat represents the in vitro production of meat without the sacrifice of animals. More specifically, it is produced from cells using tissue engineering techniques. Production primarily involves the generation of skeletal muscle tissue. However, it often includes adipocytes for fat production, fibroblasts, and/or chondrocytes for connective tissue generation and endothelial cells to provide vascularization [13].

To date, cultured meat production is a hotly debated topic (also the use of world "meat"), with conflicting opinions on its actual feasibility and sustainability. For this reason, the following review aims to discuss the in vitro meat production process by highlighting its main advantages and challenges from both a biotechnological and technical perspective.

## 2. Cultured Meat Production Process

Cultured meat production follows a precise workflow, which is briefly summarized in Figure 1.

Cell harvesting. The fundamental step in the production of cultured meat is cell procurement, for which three methods are used. (1) A cell or tissue biopsy is acquired from a living animal or recovered postmortem. The cells acquired in this way are called primary lines. (2) Pluripotent cells, such as embryonic stem cells or induced pluripotent stem cells, are used [14]. (3) Immortal cell lines.

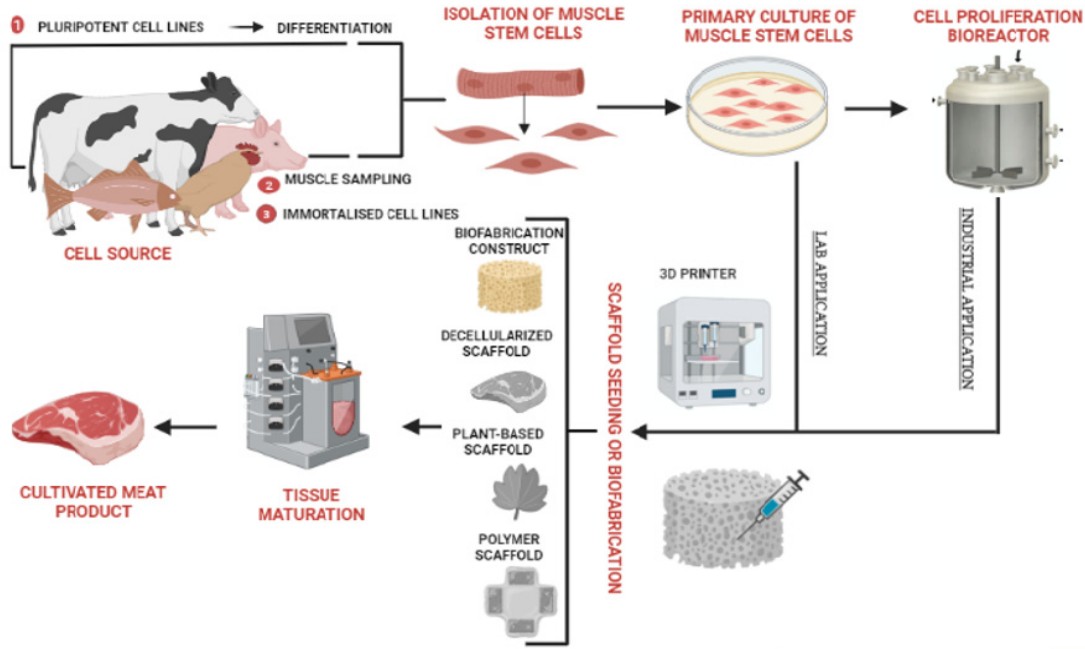

**Figure 1.** Cultured meat production process (Source Biorender, Toronto, ON, Canada).

Although primary cell cultures make it possible to study the mechanisms of cultured meat production on short time scales, they undergo a finite number of divisions before reaching senescence. Immortal or continuous cell lines, on the other hand, do not undergo senescence and can therefore undergo infinite divisions, allowing more consistent cultured meat to be obtained without the need for continuous animal biopsies. However, there are currently no immortalized cell lines suitable for cultured meat available to researchers and developers. The most similar existing cell lines are the myoblasts of model species commonly used in research, such as mice, rats, hamsters, and Japanese quails. These cell lines are characterized by the absence of taste, nutrition, and texture of traditional meat. Therefore, the continuous lines used for the production of cultured meat should be developed from cell types and species that are familiar to consumers and that are tasty, nutritious, and safe for food [15].

Working with tissue biopsies involves an additional step. It is necessary to isolate individual muscle stem cells from the rest of the fibers, achieved using proteases including trypsin, pronase, dyspase, and collagenase. After enzymatic dissociation it is crucial to remove fiber fragments, tissue debris, and connective tissues in order to ensure an efficient selection process [16].

The next step is to maintain the harvested cells in culture in order to first achieve cell proliferation and then cell differentiation and/or maturation with the ultimate goal of increasing biomass.

<u>Proliferation.</u> The cells are initially allowed to proliferate small flasks in which they can grow in two-dimensional sheets, anchored to a surface. This method represents the standard for laboratory cell culturing methods, mainly devoted to research activities. However, for large-scale industrial cell production, suspension cultures in bioreactors are required to increase the efficiency of the process: these systems require generating a great number of cells with minimal energy and resource consumption (i.e., culture medium, consumables materials), time, and handling steps [17].

The current cells culturing method involves the use of specific growth media, which contain all those substances necessary to ensure cell proliferation such as glucose, inorganic salts, amino acids, vitamins, growth factors, antibiotics, and antifungals. Other substances can be added to confer organoleptic properties, such as proteins or pigments that give a meat-like color [18]. However, to date, viable alternatives to classical culture media are

being sought, where the goal is to introduce into the medium edible by-products to ensure rapid cell growth. The same principle should be applied to fetal bovine serum (FBS), which is essential for cell proliferation, but not at all ethical and sustainable.

Scaffolding. Tissue fabrication and maturation processes can be accomplished only if the cells are provided with the correct environment in which they can adhere, proliferate, and differentiate [19,20]. Scaffolds are three-dimensional structures characterized by correct porosity, texture architecture, and mechanical and chemical properties suitable for specific cell type maturation. Moreover, considering the food engineering purposes, scaffolds must be biodegradable or edible or both, and their structure is also involved in the final organoleptic properties of the final product [20]. The traditional approach, based on tissue engineering, involves scaffold fabrication and subsequent cell seeding in a 3D culture. Scaffolds can be obtained through tissue/organ decellularization or can be fabricated. There are several different techniques for scaffold production. The most used are solvent casting and porogen leaching (SCPL), phase separation, gas foaming, sintering, electrospinning, self-assembly, and 3D printing (3DP) techniques (e.g., powder-bed 3D printing, fused deposition modelling, and stereolithography) [18].

Biofabrication. Recently, another approach is gaining importance: the biofabrication [19]. This process enables the production of cell-laden constructs using materials containing mixed cells and biological molecules. The main advantage is the possibility to fabricate thicker structures and to control the spatial arrangement of cells, even of multiple types, which are the major limitations of the conventional scaffolding methods [21]. 3D bioprinting (3DBP) is biofabrication technology based on additive manufacturing (AM) allows precise cell positioning, density deposition, and precise control of the structure, and the ratio amongst different populations in case of multicellular constructs. Cell–materials mixtures used in the process are called bioinks [22]. 3DBP represent a possible game-changer in the field of the in vitro meat manufacturing, assuring high scalability, complexity free process, minimal energy expenditure, and lower N20 emissions [22]. The major bioprinting methods are extrusion-based, inkjet, stereolithography, and laser- and light-based methods [23].

Tissue differentiation and bioreactors: Whatever the construct fabrication, cells inside the 3D scaffold must be placed in a bioreactor for maturation. Mechanical, chemical, and eventually electrical stimuli are required to complete this stage through dedicated culturing systems [24,25]. Bioreactors are closed, automated systems in which cells can proliferate, differentiate, and maturate within the construct forming the tissue. Bioreactors provide precise control over relevant variables such as temperature, oxygen concentrations, pH, and cell density [18]. Several types of bioreactors are used, the most common being static culture, spinner flask, and perfusion bioreactors (e.g., stirred tank or rotating wall) [25]. Inside the bioreactors, cells go through proliferation and differentiation. The last process is triggered by changing the scaffold and culture media and through the addition of elements such as proteins, amino acids, and minerals [24]. The lack of cell adhesion and the creation of structures suitable for cell growth are the critical points of this complex procedure.

From laboratory to industrial scale-up: The number of cells required to produce 1 kg of protein from muscle cells is in the range of $2.9 \times 10^{11}$ to $8 \times 10^{12}$ [25]. To achieve these huge numbers, it is necessary for a cell proliferation stage to occur in large-scale bioreactors [26], such that cells can grow and replicate in the order of 5000 L [27]. According to Zidaric et al., bioreactors will be essential in the industrial production process of cultured meat [26]. Tissue development would require at least two different stages: (i) the cell proliferation phase to provide a sufficient number of cells for construct fabrication (e.g., using a stirred tank bioreactor), and (ii) the tissue differentiation to a stage (e.g., a porous scaffold placed inside a perfusion bioreactor) which will lead to the final cut of meat providing the proper chemical, mechanical, electrical stimuli [26,27]. Existing products and processes in the chemical and biomedical industry do not meet requirements for large-scale cell and tissue cultures, mainly in terms of cost and sustainability. According to Specht et al., there is the perspective to develop methods and technologies to achieve the goal

of clean in vitro meat in the near future [28]. However, beyond industrial requirements, small-sized laboratory meat cultures also face several common challenges. Cells used for cultured meat production require anchor sites in order to grow and proliferate. A solution to this problem, especially within 3D printing and bioprinting, is the optimal formulation of biomaterials and bioinks, and mixing products such as polystyrene, gelatine, collagen, or protein-based additives to provide adhesion points for the cells, preventing them from remaining suspended within the construct [29]. Such anchoring sites are critical to allow cells to grow on scaffolds and structures specifically textured to ensure muscle cell differentiation. Such scaffolds are responsible for ensuring cultured meat products with multiple structural properties, including cell shape and arrangement [14]. Scaffolds can be created from many materials, which may be of natural origin (cellulose, decellularized plants, alginate, chitosan, collagen, and gelatine) or synthetic (polyethylene glycol or polyacrylamide) [18]. To date, a great concern is that animal-derived products are the most suitable and efficient for cell affinity and attachment [29], but they are inadequate in the view of an animal-free method for the cultured meat process. Moreover, since the final product must be edible, tissue scaffolds, if not edible, must be at least biodegradable and non-toxic or alternatively designed to be removable before consumption [14,18]. Once it has reached maturity, the final product must be harvested, which adds a step in the process, increasing the complexity of the procedure. This step can be performed using enzymatically, chemically, or manually, with the first two being preferred [30].

Hence, the cultured meat and its industrial process scale-up for commercial purposes is a complex and multidisciplinary matter that requires a synergic effort from the biological, chemical, technical, and industrial field. The main research focus must be the development and improvement of cell lines available to set up cell culture and culture media, bioreactors, cell lines, scaffolding, and biofabrication.

## 3. Challenges Related to Cultured Meat Production

The benefits of cultured meat production are widely discussed in the literature. Cultured meat production would lead to an 89% reduction in water used, a 99% reduction in land used, and a 96% reduction in greenhouse gases (GHGs) as a result of moving away from intensive livestock production [14,31]. While these aspects are interesting but still controversial, it is also true that there are multiple aspects about in vitro meat production that need to be considered, both from a biotechnological and technological point of view (Table 1).

**Table 1.** The main biotechnological and technical challenges for cultured meat production.

| Biotechnological Challenges | Technical Challenges |
|---|---|
| Choice of animal for cell harvesting | |
| Choice of site of collection | |
| Methods for cell harvesting | Scaffold fabrication |
| FBS: ethical challenges | Alternatives to scaffold fabrication |
| High cell proliferation and genetic instability | Biofabrication and 3D bioprinting |
| Nutritional and functional properties of cultured meat | Bioreactors |
| FBS alternatives | Industrial process scale-up |
| Food control system for cultured meat | |

### 3.1. Biotechnological Challenges

3.1.1. Choice of Animal for Cell Harvesting

The first challenge facing in cultured meat production is the choice of animal from which to perform the cell biopsy. The choice should not be random, but must take into account multiple variables, including age, sex, and rearing conditions, as they affect the presence or absence of satellite cells (adult skeletal muscle stem cells).

As the animal ages, the concentration of satellite cells within the muscle decreases. The fastest decrease in the number of satellite cells occurs most during the first few months post-

birth. In addition, because satellite cells in younger animals have undergone fewer mitotic cell divisions, they might retain their differentiation capacity for a longer proliferative period [14,31].

The sex of the animals is another factor involved in the proliferation of muscle stem cells.

Sex hormones such as estrogen and testosterone are able to influence cell growth. Compared with female animals, male animals are characterized by a higher content and activity of satellite cells, thanks to the positive influence of testosterone [16]. This beneficial action is also underlined by Mulvaney et al., who demonstrated that castrated animals presented a lower concentration of activity of satellite cells than non-castrated ones, a trend reversed with the administration of testosterone propionate [32].

Animal rearing conditions also play an important role in the composition of muscle fibers. Vestergaard et al. showed that extensively reared animals had a higher number of type I muscle fibers but also a reduction in type II muscle fibers compared to intensively reared animals. This difference, according to the authors, is due to the different diets to which the animals are subjected; more precisely, to the amount of roughage consumed, which is higher in animals reared under extensive conditions [33].

### 3.1.2. Choice of Site of Collection

Another fundamental parameter to consider concerns the biopsy site. The concentration of satellite cells varies between muscles or muscle groups. It has been shown that type I, slow-twitch fibers, are characterized by a higher number of satellite cells in contrast to type II fast-twitch, muscle fibers, in which the concentration is lower [31].

More specifically, in cattle, muscles belonging to the chuck contain primarily type I fibers, while those of the round contain predominantly type II [31].

### 3.1.3. Methods for Cell Harvesting

Another factor to be considered is the method of taking the cell biopsy. Satellite cells can be harvested either at the time of slaughter of the animal, which is not an accepted route for cellular agriculture, or through biopsies of muscle tissue. Tissue biopsy, a procedure widely used within veterinary medicine, involves the use of needle biopsy. Although this technique is quick and causes little stress to the animal, it allows a limited sample to be taken, around 0.5 g, and is not very precise due to the blind nature of the sample [31]. A second option involves a small incision at the sampling site, allowing more sampling, around 15 g, and greater success, although it is characterized by greater invasiveness for the animal [31].

To reduce the number of donors required for cultured meat production, it is desirable to maximize the number of biopsies taken from each animal, considering the levels of stress and discomfort caused. As reported by Melzener et al., a possible approach to harvesting muscle biopsies for cultured meat production could involve taking multiple biopsies (up to four) from each donor animal in one session every three months, using the needle biopsy technique, thus ensuring appropriate recovery times for animal welfare [31].

### 3.1.4. Fetal Bovine Serum: Ethical Challenges

One of the cornerstones of cellular agriculture is to ensure the sustainability of the production process. Today, almost all cell cultures involve the addition of FBS to the culture media to ensure optimal growth. Fetal Bovine Serum is an extremely complex mixture that, in addition to providing a large number of constituents such as fatty acids, lipids, vitamins, carbohydrates, inorganic salts, proteins (more than 1800), and more than 400 metabolites, provides essential hormonal factors for cell growth and proliferation. Parallel to the nutritive function, FBS ensures adhesion and diffusion factors that act as germination points for cell attachment. In addition, it allows minimizing physical damage caused by pipette handling and agitation [34,35]. Although the positive aspects of the use of FBS are widely demonstrated and discussed in the literature, it is also true that it

presents multiple problems related to sustainability and ethicality by clashing with the basic principle of cellular agriculture.

The collection of FBS has always caused a stir. When a pregnant cow is slaughtered, the fetus is removed and a cardiac puncture is made in the still-beating heart to collect serum under the most aseptic conditions possible, causing first enormous suffering and then death [34]. It is essential that fetuses are at least 3 months old to ensure anatomical formation of the heart to ensure perfect serum collection [36]. The exact amount of FBS produced and sold in the world is unknown. However, it is estimated that approximately 800 thousand liters of FBS is sold annually, which translates into approximately 2 million fetuses sacrificed [35]. However, these numbers are expected to increase exponentially due to a rising use of cell cultures for, recombinant proteins, vaccines, and therapeutic diagnostic treatments [37]. From a process sustainability perspective, the FBS market is highly dynamic, leading to continuous price fluctuation and making it unsustainable on a large scale [38]. Price and availability fluctuate due to changes in the number and cost of cattle raised worldwide, import regulations, beef and dairy prices, costs, and weather conditions [34].

FBS being an animal derivative is characterized by qualitative and quantitative differences dependent on the batch to which it belongs and therefore on the animal used for sampling, which makes it necessary to test the product before its use [39]. In addition to issues regarding variability, serum may contain varying amounts of endotoxins, hemoglobins, and other factors adverse to cell growth, as well as being potential sources of microbial contaminants such as fungi, bacteria, mycoplasma, viruses, or prions introduced during the sampling phase [34]. In addition, many other substances within FBS are still unknown, which prevents us from studying and knowing the possible effects on cell viability [40].

Therefore, it is necessary to find viable alternatives to FBS to support the large-scale production of cultured meat.

### 3.1.5. High Cell Proliferation and Genetic Instability

Cultured meat production involves the processing of cells characterized by a high proliferative capacity. However, there is always a possibility of genetic instability that can lead to the formation of cancerous cells within the culture without being clearly identified. These cells, although harmless, as they are dead at the time of meat consumption and therefore not incorporated alive inside the body, represent a great challenge of acceptance for the consumer (because the cells are subsequently digested inside the stomach), which is why this process must be further investigated and studied to ensure the total absence of risks [41].

### 3.1.6. Nutritional and Functional Properties of Cultured Meat

One of the greatest challenges for cultured meat is matching the nutritional, functional, and organoleptic properties typical of conventional meat. The ultimate goal of cultured meat production is to create a product that is as close as possible to the original one. However, to date, we are far from achieving this. As far as the consistency of the final product is concerned, it cannot be similar to that of the original meat, which becomes tender only after the animal is slaughtered, when the supply of oxygen ceases, triggering multiple biochemical changes that lead to the formation of lactic acid, responsible for the reduction of pH, which activates different families of enzymes necessary for the breakdown of muscle proteins and the subsequent tenderization of the meat. This process is referred to as maturation and, to date, it is a less considered problem in literature but not less important to be deepened with further studies [41,42]. Another discordant characteristic is the color of the final product. Meat color varies according to two basic parameters: myoglobin and iron concentration. The color of artificially produced fibers is yellow, distant from the pink/red color of the original product [41]. This discordance occurs because myoglobin is repressed by cultured cells in the presence of oxygen, and because commonly used culture media such as IMDM, RPMI1640, and DMEM contain minimal iron content. This problem can

be addressed if the media is supplemented with iron, but this supplementation remains limited [41–43]. A further problem relates to taste. Many of the biochemical metabolites present in conventional meat are net products of food intake and biological metabolism but are not derived from the muscle itself [42]. Furthermore, animal meat is the result of complex interaction of proteins, carbohydrates, flavors of the lipid fraction, nerves, and blood vessels that give the product its characteristic final taste [41]. Research groups are developing co-cultures with fat cells in order to achieve this goal, as well as to provide micronutrients such as vitamin B12, essential for human health and is easily introduced within the diet through meat consumption but would risk being lost with the production of cultured meat [41].

Over the years, intensive breeding has undergone profound changes, helping obtain safe, nutritious, quality products for the consumer.

Red meat, in fact, is considered a high source of protein which provides about 20–24 g of protein per 100 g. This value, together with the fat content, guarantees a high energy intake. The latter is the main source of energy in the human diet and its content varies according to the type of meat considered. The profile of fatty acids in red meat varies according to the proportions of lean meat and fat present. The former is richer in polyunsaturated fatty acids (PUFA), whereas fat is characterized by a high saturated fatty acids (SFA) content, containing about 37 g of SFA per 100 g of meat.

Overall, lean red meat contains similar proportions of monounsaturated fatty acids (MUFAs) and SFAs, although the exact proportions vary depending on the type of meat. The main SFAs found in red meat are palmitic acid (C16:0) (about half) and stearic acid (C18:0) (about one-third). While the former appears to increase blood cholesterol levels, the latter has a neutral effect on total and LDL cholesterol. Red meat also contains smaller amounts of myristic acid (C14:0) and lauric acid (C12:0), which are thought to raise blood cholesterol more potently than palmitic acid. In addition, although it contains low levels of PUFAs, red meat forms a substantial part of the diet, providing 18% of n-6 PUFAs (linoleic acid) and 17% of n-3 PUFAs (α-linoleic acid), contributing about 23% of total fat intake [44].

Therefore, cultured meat must include these characteristics in order to be a nutritionally competitive product. Finally, it is difficult to think that in the near future there may be a supply of in vitro meat such that consumers are offered a variety of muscles or cuts of the animals. In fact, the sensory quality of meat differs between species, breeds, genera, and types of animals, as well as the conditions under which they are raised [7].

### 3.1.7. FBS Alternatives

As previously reported, to comply with the principles of cellular agriculture, it is necessary to find reliable alternatives to FBS for cell cultivation that guarantee sustainability and ethical development.

Several studies have been performed to meet the demand for edible FBS alternative. However, most of the studies in the literature seem to be in conflict with the principles of cellular agriculture, as they applied animal-based alternatives to FBS such as fetal serum from other species (e.g., goat) or other animal by-products (e.g., bovine ocular fluid), which, although very efficient, are characterized by the same problems as FBS [35,45]. Similarly, human platelet lysate has also been considered due to its ability to promote the proliferation of stem cells derived from human adipose tissue, but being human-derived, it is not suitable for consumption [34,46].

When working with cell culture, the use of chemically defined media (recombinant protein and growth factors) is a common practice. These, although ethical and suitable for human consumption, are characterized by a high cost that makes them unsuitable for large-scale application.

It is therefore essential to study innovative matrices that, when added to the culture medium, can sustain cell proliferation and viability in both the short and long term, thus ensuring the sustainability and ethicality of the production process.

Possible matrices tested on the proliferation of different cell culture are those reported by Ho et al. [47] (Table 2). However, although most of them may be good candidates as alternative to FBS due to their high proliferative capacity on cell, they need to be discussed in terms of sustainability and ethics.

**Table 2.** Applications and analysis of different matrices in cell cultures. Modified by Ho et al. [47].

| Matrices | Cell Type | Effects | Refs. |
|---|---|---|---|
| **Plant peptones** | CHO-320 (CHO K1 clone) | Improved cultivation and productivity of Human interferon gamma | [48] |
| **Yeast hydrolysate** | CHO rCHO (recombinant CHO) | Higher productivity of Human beta interferon Higher cell growth | [49,50] |
| **Rice protein hydrolysate** | CHO-320 Human HepG cells | Protection against oxidation stress from hydrogen peroxide | [51,52] |
| **Soy peptones** | CHO DG44 | Improved cell production | [53] |
| **Wheat hydrolysates** | CHO | Improved cell viability | [54] |
| **Marine cyanobacterium Spirulina *maxima*** | Human Lung Carcinoma | Improved cell viability and proliferation | [55] |
| **Chlorella vulgaris extract** | CHO-K1 and MSC | Promoted cell growth | [56] |
| **Rapeseed caked** | CHO-C5 | Promoted cell growth | [57] |
| **Silk sericin hydrolysate** | CHO and Hela cells | Improved cell growth and proliferation | [58] |
| **Whey protein** | CHO K1 JURKAT E6.1 | Improved cell viability and proliferation | [59] |

The matrices shown in Table 2 are those that achieved promising results in cell proliferation and viability. Most of these are of plant origin (plant peptones, rice, soy wheat, Marine cyanobacterium Spirulina *maxima*, Chlorella vulgaris, and rapeseed), in light with the principle of cellular agriculture. Particular attention should be focused on hydrolysates. Their relatively low cost makes them very attractive as FBS replacement components. However, as hydrolysate products are not fully characterized, further understanding of their components and the mechanism by which they influence cell growth and maintenance is crucial to their large-scale application [47].

Sericin is a macromolecular, globular, biodegradable, and biocompatible protein produced within the central gland of silkworms. It is obtained by degumming the cocoon of *Bombyx mori*. For a long time it was considered a waste product of silk processing until researchers explored its potential within pharmaceutical, biomedical, and cosmetic application [35,60]. In addition to these fields, sericin finds application as an edible coating material within food products due to its ability to retard oxidative activity to polyphenols damage, opening an interesting debate on its possible use as an alternative to FBS in the maintenance of cell culture [61]. Being an animal-derived product, its use would not fully satisfy the requirements of ethics and edibility. However, it must be reiterated that this is a waste product of the silk-processing industry and its use within a different sector can meet the circular economy principle.

Whey protein is one of the main components of milk. Specifically, it is a by-product of the dairy industry characterized by a high biological value thanks to its antioxidant, anti-inflammatory, antiviral, and antitumor properties, expressed both when consumed individually and as an additive in other foods [62–64]. These functions can be attributed to its excellent nutritional composition; it consists mainly of $\alpha$-lactalbumin, albumin, $\beta$-galactoglobulin, and immunoglobulin [62]. For these reasons, it has been investigated as a substitute for FBS within culture mediums.

Again, this is a product of animal origin. However, it is a waste product with a high environmental pollution load as reported by Veskoukis et al. [62]. Indeed, it has been calculated that its polluting potential is equal to a biochemical oxygen demand

approximately 175-fold higher than the sewage system of modern cities. Therefore, it causes serious environmental problems when discarded [62].

For this reason, although it is of animal origin, its use as an alternative to FBS might be considered. Moreover, its production is high; this would make it possible to meet the high market demand typical of FBS.

Overall, it is of paramount importance to find suitable alternatives for FBS, considering the combined use of plant-based products (e.g., those reported in Table 2). However, due to the high variability in their chemical composition and mechanism of action of these matrices, it is necessary to investigate the effects of their combination on cell cultures.

### 3.1.8. Food Control System for Cultured Meat

A fundamental aspect within any production is monitoring along the entire production chain. As reported by Chriki and Hocquette, there has been much discussion about the safety standards of cultured meat. Proponents of in vitro meat consider it a much safer product than a conventional one, due to the fact that it is produced in a closed and controlled environment with no possible contact with external pathogens. This aspect plays a key role, especially during the slaughter process of the animal, where pathogenic intestinal bacteria such as *E. coli*, *Salmonella*, or *Campylobacter* can contaminate meat which is subsequently marketed [7]. However, the objective of completely eliminating possible risks throughout the production chain is difficult to achieve, and so it is necessary to adopt appropriate controls to identify these risks before marketing the product. At the same time, it is indisputable that, although food disease episodes occur, every year the surveillance of the meat production chain increases its quality standards, ensuring increasingly safe products. Therefore, production in a closed and controlled environment, combined with surveillance along the entire supply chain, characteristic of conventional production, would make cultured meat production a possible safe product.

## 4. Technical Challenges in Cultured Meat Production

The main technical challenge in producing cultured meat is replicating the 3D environment of real muscles, in which cells can maturate in a laboratory or a factory to mimic the tissue. This complex process involves a great number of tasks and unsolved problems, which can be globally aggregated at a higher hierarchical level into three main categories: scaffolds, biomaterials, or bio-inks and their interaction with cells, fabrication procedures, and culture processes for cell proliferation and differentiation techniques [65,66]. From a technical point of view, well-established methods and processes from tissue engineering and regenerative medicine are used as after adapted to the specific purpose [67]. In the following sections, conventional scaffolding to cutting-edge technologies, such as bioprinting, are presented, in terms of their current state and perspectives. Moreover, the most common bioreactors as culture systems are described.

### 4.1. Scaffold Fabrication

A scaffold can have a porous, tubular, or tissue-like structure. The most important parameters are porosity and material composition. The type and structure of the scaffold depend on the specific application for which it is designed.

However, the general requirements to be fulfilled are essentially to allow cells to adhere and allow material transport through its structure.

The following sections present several methods for fabricating porous scaffold, divided into conventional (Figure 2) and non-conventional fabrication techniques.

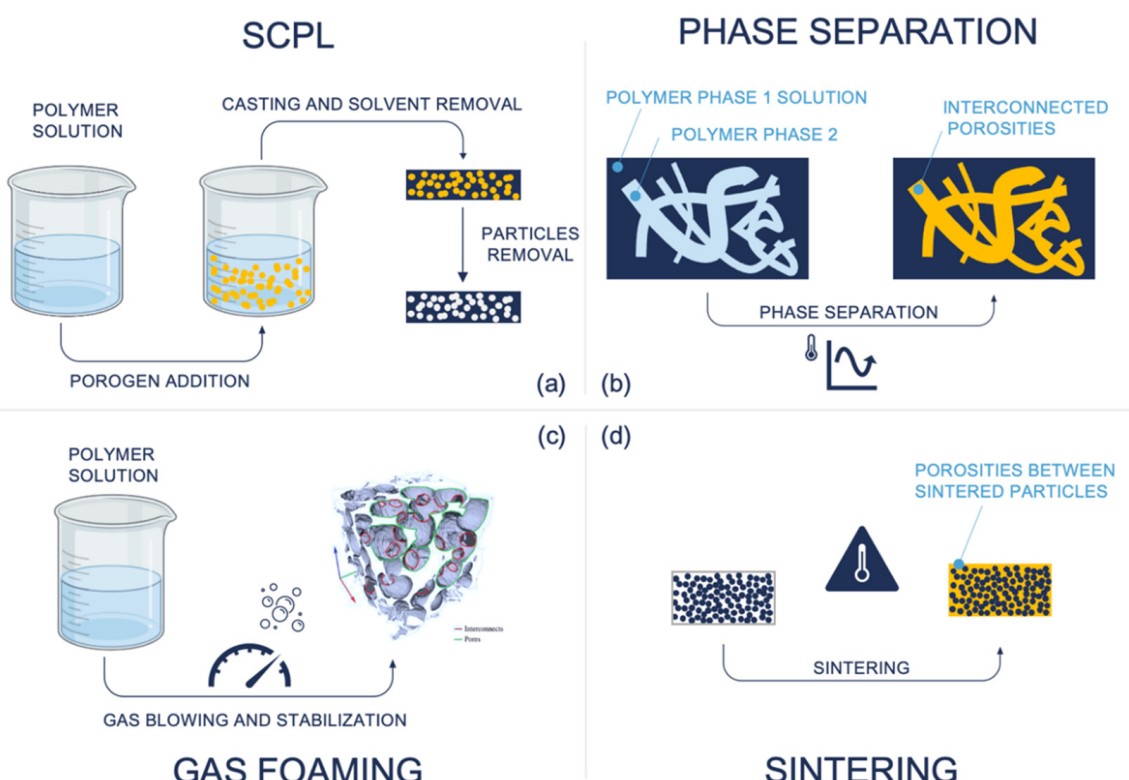

**Figure 2.** Conventional porous scaffold fabrication technologies process representation and example of scaffold, respectively (**a**) SCPL, (**b**) phase separation, (**c**) gas foaming, (**d**) sintering.

4.1.1. Conventional Porous Scaffold Fabrication Technologies

- **Solvent casting and porogen leaching** (**SCPL**) [68]: This process involves mixing a polymer solution dissolved in an organic solvent composed of insoluble particulates (porogen). The mixture is then cast into a mold or a membrane, and the solvent is evaporated. Finally, the structure is immersed in an aqueous solution to leach out particulates in the structure. Porosity, in terms of shape, size, and uniformity, depends on the particulate choice of particulates, typically salt particles. The main drawbacks are the lack of control of internal architecture and uniformity, reduced reproducibility, formation of a skin layer due to polymer thickening that can limit access to internal porous, limited thickness (2–3 mm) [69], weak mechanical properties, and possible cytotoxicity due to residual solvent and porogen [70].

- **Phase separation** [71]: The technique is used to produce a scaffold through the separation of a mixture into two phases: a polymer-rich one and a polymer-poor one. This is achieved under thermodynamically unstable conditions. For example, cooling the solution below the freezing point of the solver induces crystal nucleation inside the solution; after that, the solid material is sublimed, ensuring that the structure is composed of only the polymer-poor region with porosity, because the solvent and the polymer-rich phase are evacuated from the scaffold. This technique leads to highly interconnected porosity which can be used to reproduce channel-like structures by applying a directional temperature gradient. Nevertheless, control and optimization of process parameters (e.g., temperature, polymer concentration, surfactants use, crystal nucleation) are the main problems in managing pore size and distribution [72]. Moreover, the typical pore size achievable is often smaller than the typical dimension in tissue engineering applications (<200 μm).

- **Gas foaming** [73]: This is a class of techniques for scaffold fabrication exploiting a blowing agent to generate gas inside the material which acts as a porogen agent. The main advantage is the absence of solvents or porogen materials, which can induce

cytotoxicity due to possible residuals. The Gas formation can be induced chemically or thermally or by pressure change. The main drawbacks of the technique are low control over pore size and interconnectivity, low reproducibility and structural uniformity, and difficulty in incorporating biological molecules in thermally induced processes [74,75].

- **Sintering** [76]: The technique is used to produce cohesive porous scaffolds through bonding of a polymeric phase and ceramic particles or fibers. The usual procedures involve a bed of randomly packed particles bonded through heating up to a temperature above the glass transition temperature of the base material, but lower than its melting point, creating a local fuse area only in the contact surfaces, leading to a porous microstructure. Alternative sintering modes are mild solvent treatment and pressure. Sintered scaffolds are characterized by lower porosity, small pore size with difficulty in precise control and distribution, and higher mechanical properties, and they are mainly used in dental and bone-repairing applications [77].

4.1.2. Non-Conventional Porous Scaffold Fabrication Technologies

- **Electrospinning** [78]: The method is based on an electric field generated between a polymer solution delivery system at a controlled flow rate and a collector, drawing the solution into a fiber, an illustrative example is shown in Figure 3. The result is a membrane of non-woven fibers. The textile-based technique has been created to reproduce fiber-based materials, such as those similar to the extracellular matrix (ECM). The resulting porosity is interconnected and the achievable pore size is lower than that by other scaffolding techniques, achieving fibers with diameter up to a few nanometers [78], which can be an advantage for specific applications (e.g., vascular [79]), but tends to limits the cell migration to a point where its applicability in tissue engineering becomes a problem. Several process parameters can be controlled to tune fiber diameter and alignment, adapting textural properties to the specific cell type to be seeded.

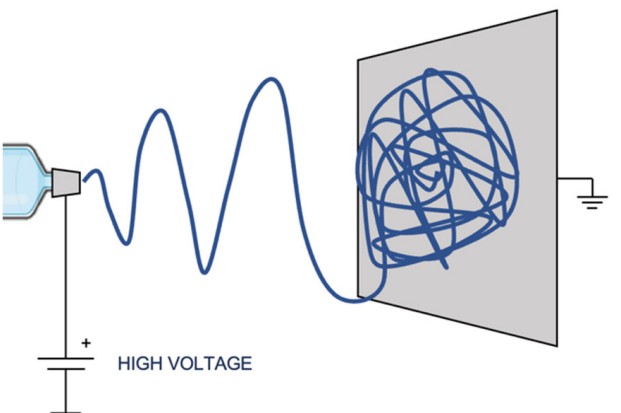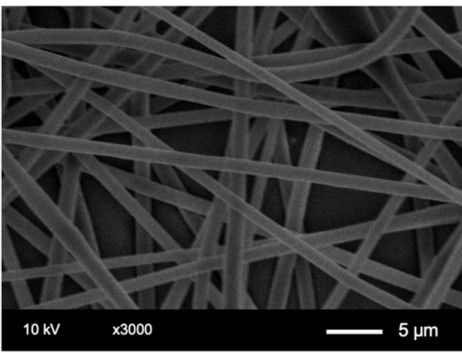

**Figure 3.** Illustrative example of the electrospinning method for scaffold fabrication and image of a construct (adapted from [80]).

- **Self-assembly** [81]: The technique involves specifically designed amphiphilic peptides with the capacity to spontaneously organize into ordered structures, including nanofibers. The method allows a great process control starting from the building blocks in a bottom-up design approach for tissue engineering application.
- **Hybrid scaffolds:** With the aim of controlling structural and composition features, mainly porosity, at different length-scales, several approaches with mixed techniques are used, such as SCPL and electrospinning combination [82], multilayer electrospun composites with different parameters [83], and a combination of more than two fabrication techniques [84,85].

- **Additive manufacturing** (**AM**): The conventional techniques of porous scaffolds fabrication, as well as other emerging alternatives, are implemented to produce scaffolds to recreate the complex micro and macrostructures of biological tissues. However, all of them have limitations and allow narrow control over important textural parameters such as pore shape, dimensions, and interconnectivity [86]. An emerging family of technologies, based on additive AM techniques, has proved to enable the manufacture and control of complex shapes. AM, popularly called 3D printing, is a generic definition and represents a large group of processes that can be classified in several ways [87]. Within the porous scaffold fabrication context, according to Rey et al., some of the AM technologies could be used to produce scaffold with a high spatial resolution, the structural complexity, and control over the internal pore architecture. The most promising techniques are powder-bed based 3DP, such as selective laser sintering (SLS), and liquid raw-material-based 3DP, such as stereolithography (SLA) [86].

### 4.2. Alternatives to Scaffold Fabrication

#### 4.2.1. Tissue Decellularization

An alternative approach to engineered scaffold is plant or tissue/organ decellularization. This approach is based on the removal of resident cells and a large proportion of the major histocompatibility complex from a tissue in order to obtain a natural scaffold to be seeded. In this way, the ECM structure is preserved [87]. There are several successful examples in tissue engineering regeneration that follow the so-called "like-to-like" strategy, in which donor and regenerated tissues are of the same type [86]. Nevertheless, in the case of tissue/organ decellularization, the tissue procurement would require the use of animal-derived tissues, completely in contrast with the fundamental prerequisite of cultured meat.

#### 4.2.2. Microcarrier Cultures

Microcarriers are beads with typical dimensions between 100 and 200 μm, and they represent a possible solution because mammalian cells require a surface on which they can grow [17]. For tissue engineering applications, especially for food products, microcarrier-based systems are retained as the major culturing system to achieve a high volume of cells because they can provide a large surface area per unit volume of medium [88]. Verbruggen et al. proposed a myoblast cell production system based on microcarriers suspended within cells and medium inside a stirred tank bioreactor, achieving promising results in terms of cell growth for efficient and cost-effective development of cultured meat [88]. Bodiou et al. provided three scenarios based on cultures with microcarriers: temporary microcarriers culture for proliferation, non-edible but degradable microcarriers, and edible microcarriers embedded in the final product. According to the author, the third is the most promising for cultured meat production [89]. Beyond the great possibilities, the main drawback of the use of microcarriers or aggregates is that the cells may form clusters that do not proliferate in the correct manner, because of which, if not modified, the cell proliferation phase would be difficult to control [17].

### 4.3. Biofabrication and 3D Bioprinting

Biofabrication refers to the production of complex biological products combining cells, matrixes, biomaterials, and biomolecules, especially for tissue engineering, regenerative medicine, and food engineering. This emerging field has been highly stimulated by the development of AM-based technologies [90].

Direct 3D printing of biological material, including cells, is defined as 3D bioprinting. The main challenge is to adapt technologies developed to molten plastics and metals to work with sensitive, soft, and biological materials (bioinks). The central objective is to reproduce the complex micro-architecture of the ECM better than other methods and have higher control on cell density and deposition [91]. The main drawback of the scaffolding techniques biofabrication aims to overcome is the limited cell migration inside porous

scaffolds. According to Sachlos et al., cells do not necessarily recognize the scaffold surface and, most importantly, they do not migrate more than 500 μm from the surface [92]. There are several 3DBP strategies, characterized by different features, used to biofabricate 3D cell-laden structures. The most diffused methods are extrusion, inkjet, and stereolithogaphy-based bioprinting, while hydrogels are commonly used as bioink base materials [93].

### 4.3.1. 3D Bioprinting Strategies

There are different bioprinting strategies, each with its own pros and cons (provided in Table 3. Figure 4 provides a graphical representation of the most common methods or those of higher research). According to Vijayavenkataraman et al., no one method could be excursively used to achieve the goal of biofabrication of complex tissue, the current trend being research in the development of hybrid methods [94].

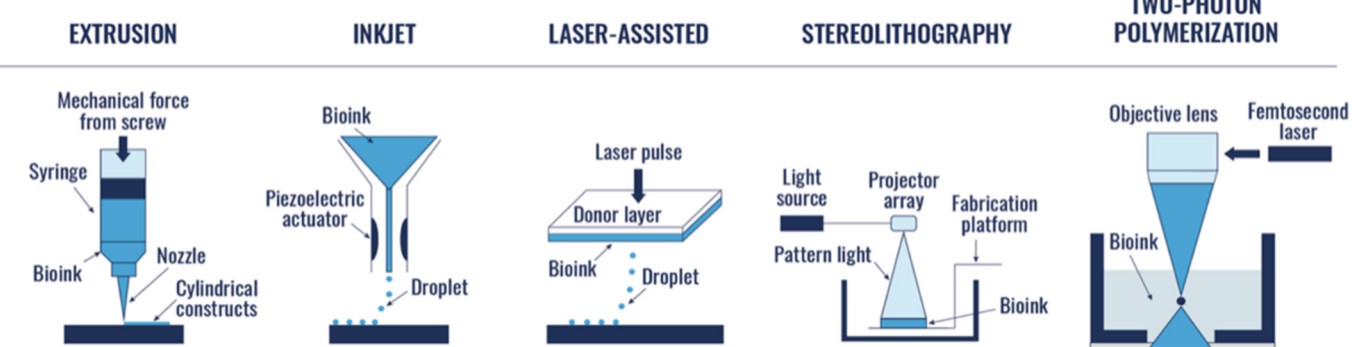

**Figure 4.** Different 3D bioprinting techniques, adapted from Santoni et al. [95].

Extrusion bioprinting: The most common 3DBP method is the extrusion-based one, mainly because it is versatile and affordable, and several entry-level bioprinters are available on the market [95]. Kang et al., for example, successfully used it to bioprint meat-like constructs [96]. Extrusion bioprinting usually relies on a dispensing system, such as a syringe with an appropriate nozzle or needle, installed on a print-head used to precisely deposit bioinks on a print bed. The dispensing system can be pressure, piston, or screw driven: the first allows better cell viability, but with low control on material flow rate and shape fidelity [91,93]. The main advantages of the extrusion-based method are scalability, printability of a wide range of materials of high viscosity, and high cell concentration [94]. Nevertheless, the resolution is the lowest if compared with the other methods and it is related to the nozzle diameter, in that reduction is limited by the consequent cell viability drop [94]. Moreover, the post-printing cell viability depends on the bioink viscosity and cell concentration [97]. Further drawbacks are nozzle-clogging and the limitation due to material rheology: only bioinks with shear-thinning property can be used [94].

Inkjet bioprinting: A liquid droplet dispensing system, based on temperature or piezo-electric driving technology is used. Due to the nature of the process, inkjet bioprinting is characterized by high speed and resolution, and associated to lower costs. However, it can be used only with low viscosity materials, and it is strongly limited by the nozzle-clogging mechanism. Moreover, to enhance and facilitate droplet formation, low cell concentration is allowed [91,94].

Stereolithography and two-photon polymerization-based bioprinting. Stereolithography is based on the polymerization of light-sensitive polymers; UV or visible lights can be used to photo-cure the material in a layer-by-layer manner [23]. It is a nozzle-free method, and so does not face the problem of nozzle clogging, which represents a great limitation in the previously explained methods. Moreover, it allows obtaining very high resolution and high printing velocity. However, only light-sensitive polymers can be used, and the UV lights, as well as UV-activated photo-initiators, can damage cells and consistently reduce post-printing viability [23,94,97,98]. On the basis of this technology, digital

light processing (DLP) has been developed to photo-cure polymers. The method uses a projector emitting visible light to overcome problems due to UV irradiation [98]. With the advent of two-photon polymerization-based stereolithography nanoscale resolution is achieved. The highest spatial resolution is among biofabrication techniques but with costly systems [94,99].

Laser-assisted bioprinting. The method is based on the laser-induced transfer principle [100]: it uses a pulsed laser beam that acts, through a focusing system, on a ribbon composed by a donor transport support material covered by a laser-energy-absorbing and a biological material layer. The focused laser pulses on the absorbing layer generates high-pressure bubbles that propel the material against a collector substrate facing the ribbon. LAB is a nozzle-free technology capable of high speed and high cell density bioprinting. However, it is less common than inkjet and extrusion, mainly due to the costs of the technology and the difficulty to produce ribbons. Moreover, it is difficult to scale-up for high-volume production [91].

**Table 3.** Comparison of major bioprinting methods considering several properties, advantages and drawbacks, and main applications. Table adapted from [23,94,95].

| Properties | Extrusion | Inkjet | Laser-Assisted | Stereolithography | Two-Photon |
|---|---|---|---|---|---|
| Speed | Slow | Fast | Medium | Fast | Very fast |
| Cost | Moderate | Low | High | Low | Very high |
| Cell viability | 85–95% | 80–95% | <85% | 25–90% | >80% |
| Cell density | High | Low | Medium | Medium | Medium |
| Scalability | High | High | Low | Medium-high | - |
| Resolution | 100–500 μm | 100–500 μm | 20–100 μm | 20–100 μm | 0.1–10 μm |
| Bioink viscosity | $6$–$30 \times 10^7$ mPa·s | <10 mPa·s | 1–300 mPa·s | No limitation | No limitation |
| Advantages | Is Simple, is capable of printing various biomaterials, high cell densities | Has the ability to print low viscosity biomaterials, fast fabrication speed, low cost, and a high resolution | Has a high resolution, is nozzle-free, and can deposit biomaterials in solid or liquid phase | Is nozzle-free, has high complexity, and has a high resolution | Is nozzle-free, has a high complexity, has the highest resolution, and has high cell viability |
| Drawbacks | Only for viscous liquids, resolution | Limited to low viscous fluids, resolution, cell density | High cost, thermal damage due to nanosecond/femtosecond laser irritation, scalability, cost | Lack of printing multi-cells, cell damage during photo-curing | Lack of printing multi-cells, cost |
| Applications | Tissue models for cell research, drug testing and regenerative medicine, meat-analogue constructs [101] | Supplementary to other technologies | Precise cells deposition | Scaffolds and complex structures with channels | Vascularized and high precision models |

### 4.3.2. Bioink Formulation

Bioink formulation is one of the most important branches in bioprinting research, with about 25% of the overall publications in the field [95]. These materials are typically based on cytocompatible hydrogels, and they must have several key properties, both from a mechanical (e.g., viscosity, printability, and stiffness) and biological (e.g., cytocompatibility and cell–material interaction) point of view [101].

Hydrogels are a class of hydrophilic polymers that can be cross-linked, forming a 3D network capable of absorbing and holding a large quantity of water (even upward of 100 times their dry weight). This property allows the polymers to reach the hydration levels found in most tissues. Moreover, the porous network allows for a great degree of

diffusion of nutrients and wastes out of the material. These hydrogels can be chemically stable, or they may degrade and dissolve [102]. Hence, they offer the opportunity to recreate engineered microenvironments suitable for cells, mimicking the natural ECM properties and native cellular niche important for tissue regeneration [86]. Hydrogels can be formed using natural biopolymers, synthetic biopolymers, or combinations of the two. The most common have been formed using proteins (such as collagen, elastin, and fibrin), polysaccharides (alginate, agarose, chitin/chitosan, etc.), and synthetic polymers (such as polyethylene glycol, polyvinyl alcohol, polyacrylamide, and polylactic acid) [86]. A more complete framework of materials used for hydrogel fabrication is provided in Table 4.

**Table 4.** Hydrophilic polymers used to synthesize hydrogel matrices, adapted from [102].

| Natural Polymers | Synthetic Polymers | Combination of Natural and Synthetic Polymers |
|---|---|---|
| Anionic polymers: HA, alginic acid, pectin, carrageenan, chondroitin sulfate, dextran sulfate | Polyesters: PEG-PLA-PEG, PEG-PLGA-PEG, PEG-PCL-PEG, PLA-PEG-PLA, PHB, P(PF-co-EG) $\pm$ acrylates, P(PEG/PBO terephthalate) | P(PEG-co-peptides), alginate-g-(PEO-PPO-PEO), P(PLGA-co-serine), collagen-acrylate, alginate-acrylate, P(HPMA-g-peptide), P(HEMA/Matrigel®), HA-g-NIPAAm, GelMA |
| Cationic polymers: chitosan, polylysine | Other polymers: PEG-bis-(PLA-acrylate), PEG $\pm$ CDs, PEG-g-P(AAm-co-Vamine), PAAm, P(NIPAAm-co-AAc), P(NIPAAm-co-EMA), PVAc/PVA, PNVP, P(MMA-co-HEMA), P(AN-co-allyl sulfo- nate), P(biscarboxy-phenoxy-phosphazene), P(GEMA-sulfate) | |
| Amphipathic polymers: collagen (and gelatin), carboxymethyl chitin, fibrin | | |
| Neutral polymers: dextran, agarose, pullulan | | |

The main advantage of natural polymers is the higher cytocompatibility and presence of recognizable biological moieties (typically only from animal sources) that can act as signals and modulate cellular responses such as attachment, proliferation, and differentiation. However, they are affected by batch-to-batch variability, they often require stringent extraction and purification protocols, and they face procurement problems related to sustainability and availability [27,86]. Moreover, those from animal-derived source are useful just for research activities, but they are inadequate for large-scale cultured meat production. Due to their source, synthetic polymers potentially can reach higher reproducibility and uniformity in mechanical and rheological behaviors, with high controllability of physical properties. In addition to these advantages, they have the worst biological behavior, lacking in moieties to interact with cells and create an appropriate environment. Thus, several combinations have been formulated to combine the properties of the two classes. Alternatively, functionalization processes are applied to synthetic polymers in order to enhance cell adhesion [86,102]. As an example, Chaudhuri et al. show the beneficial effect of alginate modification with the peptide motif RGD (arginine-glycin-aspartic acid), an integrin-binding ligand [103]. Integrins are transmembrane receptors of cells and turned out to be fundamental in tissue engineering because they activate signal transduction and regulate the cell cycle, including cell spreading, migration, guidance, proliferation, and apoptosis [103,104].

The network of the hydrogel is formed via cross-linking (fixation or gelation) with hydrogel precursor polymers. This can be carried out before, during, or after the 3D printing and it is fundamental to preserving shape and structural integrity and avoiding collapse. The gelation mechanisms can be divided into two main categories: chemical and physical. Typically, physical cross-linking is a reversible process but associated with poor mechanical stability, while chemical reagents are able to increase the mechanical stability by creating covalent crosslinks [101]. Crosslinking can be stimulated by light (i.e., UV or visible), heat, or crosslinker bath (i.e., ionic crosslinking).

A combination of mechanisms or steps may be pursued to improve the process. For example, Colosi et al. proposed a bioink blend composed of alginate and gelatin methacroyl (GelMA) at low concentrations (<5% w/v) characterized by low viscosity, which positively impacts on cell viability during the extrusion process. The ink is crosslinked in two steps, CaCl2 during (using a coaxial nozzle) and after bioprinting, and then the construct is further stabilized by UV-crosslinking [105].

The formulation and preparation of a bioink and its bioprinting is complicated because of the presence of cells and their strict requirements for sterility and viability. Concerning printability, the most important physio-chemical parameters of a hydrogel include the rheological behavior, swelling properties, surface tension, gelation properties, and kinetics. These properties must be tuned, taking into consideration the bioprinting technique (e.g., extrusion or inkjet) and the type of cell to be used. Thus, the characteristics of the bioink should meet the mechanical requirements from the process point of view and at the same time ensure cell survival after the bioprinting and within the constructs [101]. Most of the recent papers outline the necessity to find the best compromise between printability and specialization for the specific cell type or tissue under analysis [95].

According to Rutz et al., the optimal formulation of a bioink must take into account the overall process. The author refers to the extrusion bioprinting, which is to date the most common, and lists the major factors responsible for achieving an optimal bioink design capable of reaching high cell viability and evaluates their impact [106] (Figure 5).

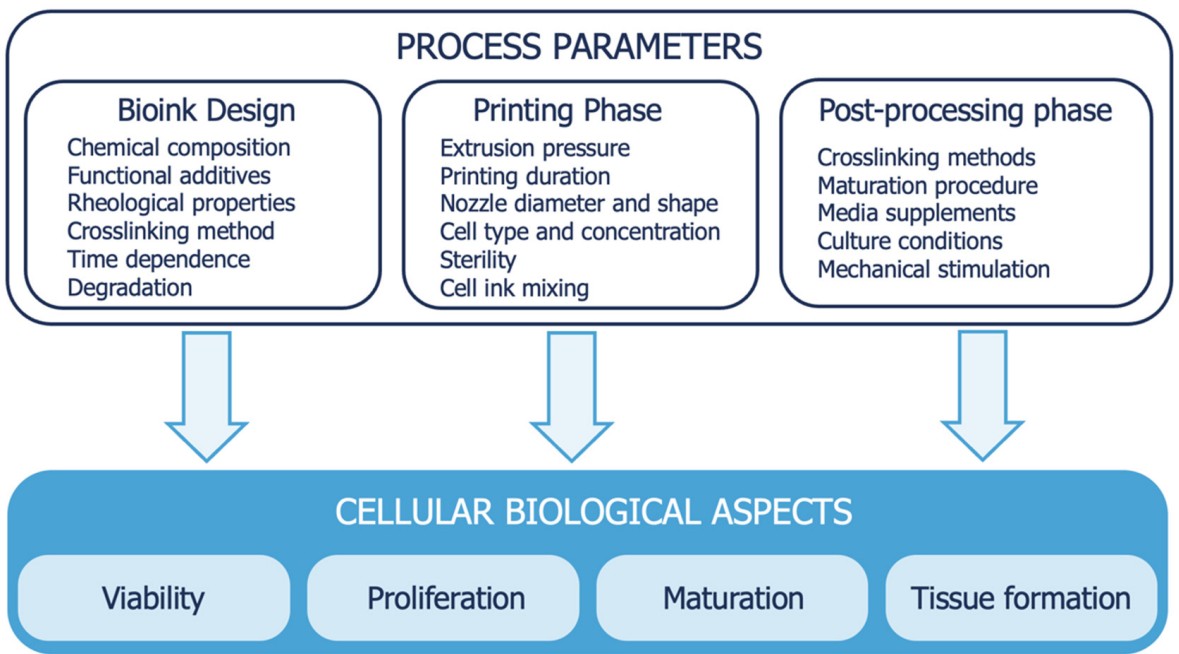

**Figure 5.** Schematic representation of major process parameters, grouped by bioink, printing, and post-printing design phases, and the biological behavior of cells on which parameters can act. Adapted from [106].

Viability is considered the key challenge because it impacts the subsequent cellular events, such as proliferation, differentiation, and tissue formation, even if many factors impacting cell stress and their severity are not yet completely understood.

Additionally, cellular density, cellular projections, and network formation retained are of fundamental importance [106].

According to Rutz et al., the major factors in bioink design affecting cell viability and network formation are the following. Their effect is schematically displayed in Table 5.

**Table 5.** Relationship between bioink material properties and cellular viability and behavior, adapted from [106].

| Bioink | Cells |
|---|---|
| Printing pressure ↑ | ↓ Viability |
| Nozzle diameter ↓ | ↓ Viability |
| Printing time ↑ | ↓ Viability |
| Degree of crosslinking ↓ | ↑ Density in bioink |
| Viscosity ↓ | ↑ Density in bioink |

- There is a circular relationship between cells and bioink rheology: the firsts impact the rheology, and thus process parameters, and vice versa. For example, Billiet et al. found a twofold lowering in the viscosity of a GelMA-based bioink when prepared with 0.5 and 1.5 million cells/mL and a fourfold lowering when prepared with 2.5 million cells/mL [107]. Hence, there is the need to predict or test the rheological properties of ink with cells inside.
- Mechanical stresses must be minimized, reducing printing pressure and increasing nozzle diameter because cells are mechanical sensing and suffer higher mechanical stresses.
- The modulus of the gel-phase highly impacts cell viability, and probably also molecular weight and polydispersity. The mechanical properties of the material surrounding the cells is a crucial aspect that is poorly understood.
- Post-printing crosslinking can affect cell viability as well. Frequently, bioprinted constructs are UV-crosslinked and the amount irradiation cells can tolerate is not clear, probably between 30 s and a few minutes. Moreover, the final degree of cross-linking can interfere with cell projections and network formation, an important mechanism to ensure tissue formation. This can become important when polymer concentration is increased to increase printability. According to the author, the concentration of the polymer must be between 5 and 10%, but it is obviously a polymer-dependent quantity.

Another important matter of focus these days is the assessment techniques to standard-ize the evaluation of printability [108,109]. Paxton et al. proposed a printability assessment method to evaluate the rheological properties of bioinks for extrusion processes. It is divided in two steps: (1) a qualitative screening of fiber formation and layer stacking capabilities, and (2) a rheological evaluation focused on flow initiation, shear-thinning, and post-printing recovery properties [108].

Through image analysis during bioprinting, other parameters related to the shape fidelity can be calculated, such as the printability index (Pr) based on the circularity calculation of a grid-like structure [109].

Research for novel solutions in bioink development for 3D cell culture and bioprinting purposes can push emerging technologies in tissue engineering and food bioprinting.

The challenge is to accomplish the correct balance between a material's chemical, morphological, and structural features that have a positive effect on cellular cycle processes. Moreover, non-replicative materials should be avoided for the standardization of the pro-duction process. In the case of re-usable materials after maturation, animal-derived material can also be used [65,88]. Specifically for the food bioprinting field, further considerations must be made. According to Post et al., the most important new requirements for edible bioinks are related to biological and environmental concerns: sustainability (i.e., water and land consumption, and energy and carbon footprint), source of raw materials (consistent, animal-free, and scalable), taste, and safety for human consumption [27].

### 4.4. Bioreactors

In the context of tissue engineering, bioreactors are used to apply and control envi-ronmental parameters and conditions to constructs or cell cultures. The most important parameters in culture are temperature, pH, $CO_2$, and other biological, biochemical (such

as oxygen, nutrients transfer, or waste removal), and physical (mechanical stimuli) conditions [25]. The specific requirements and the architecture of a bioreactor depend on the cell type or tissue culture. Thus, they must be designed and manufactured for tissue-specific purposes [25].

- **Static culture systems:** They are the simplest and provide the required nutrient in a static fluid environment. Thus, the media must be changed often, and it perfuses by passive fluid diffusion [25]. These systems can be easily coupled with load-bearing mechanisms, for example, to provide a compressive load to the engineered tissues [110,111].
- **Spinner flasks:** Spinner-flask-based systems are used to apply fluid-induced shear stresses to constructs submerged within a re-circulating and nutrient-rich medium solution [25]. Although this system provides a better environment to construct with respect to the static culture, spinner flasks may not be optimal due to turbulent flow and the related higher shear stress generated [112].
- **Perfusion systems:** The poor diffusion condition of static culture can be improved by perfusion bioreactors, especially in the internal parts of porous scaffolds [113]. These systems are characterized by a culture encasing bioreactor, vessels for the medium (nutrient-rich and oxygenated), and a pump generating the flow [27]. Moreover, perfusion systems allow for automatic media circulation, waste removal, and provide shear stress due to the flow, which is beneficial in specific tissue cultures such as for dermis and cartilaginous tissues [23,112].
- **Rotating wall vessel:** An alternative approach for reducing diffusional limitations of nutrients and waste with limited shear stress is the use of rotating wall vessel bioreactors [24]. Although shear stress is important for cell maturation, an excessive force will lead to damages or to the formation of undesired capsules surrounding the tissue [112]. This method uses a dynamic laminar flow induced by the rotating fluid inside the bioreactor, and it has been proved to be effective for cell cultures, especially chondrocytes and cardiac cells [24]. The main drawback is the non-uniform tissue growth, due to the force field. Moreover, the centrifugal force can cause collision between scaffolds and the walls of the bioreactor [112].
- **Pulsatile flow:** For cardiovascular cell cultures that require a pulsatile stimulation, bioreactors exploiting pulsatile flow are used to mimic in vivo conditions. Typically, vascular cells are cultured into tubular scaffolds [112].

With technological and design tools, increasingly powerful bioreactors designed for specific application can be fabricated, with high specialization and effectiveness [24]. The main assumption in bioreactor design is that the same factors and stimuli that determine phenotypic nature and functionalities to tissue and cells in vivo also determine the progression of cells in vitro [114]. Several tissues are cultured, providing mechanical, electrical, chemical, and mixed systems mimicking the environment of in vivo conditions. Skin and cartilaginous tissues are one of the most successfully cultured—the last mainly due to its avascular nature—providing mechanical stimulation within static or dynamic cultures [23,115]. Additionally, high shear-induced flow is used to culture bone tissue [116]. Zimmermann et al. implemented a system capable of applying passive cyclic mechanical stretch inside the culturing system to constructs for cardiac tissue engineering [117,118]. For cardiac tissue, perfusion bioreactors capable of also providing electrical stimulation are specifically designed and tested [119–121]. Other specific bioreactors are implemented for heart valves and blood vessels [112,120].

*4.5. Industrial Process Scaling-Up*

In cultured meat production, there are several technological challenges to reach an appropriate scaling-up. Within this context, the key issue is related to large-scale bioreactors for a high volume of cell production and tissue maturation. According to Post et al., the research will test other configurations and bioreactor types to achieve higher cell densities by minimizing bioresources utilization and costs to make cultured meat a commodity [27]. According to the same author, the initial cell production and tissue maturation will be

two separate and different stages with different problems involved. The first is related to cell proliferation by a suitable multiplier factor—not lower than $\times 10^9$—and aims to maintain cells in an exponential growth state and prevent them from undergoing precocious differentiation. The second is related to providing correct stimuli and providing nutrients in an efficient way [88]. The industry standard for mammalian cell bioreactors is stirred tanks, in which cells are in suspension, aggregated, or attached to microcarriers [21,31]. To establish efficient and cost-effective microcarriers-based culture systems, several challenges must be faced, the first being the surface and physical properties of the microcarriers: charge, coating, surface, and size [88]. Moreover, according to the outlook provided by Bodiou et al., the relationship between the microcarriers and the final product must be addressed, considering technological possibilities. The temporary microcarriers scenario presents the unsolved problem of cell separation and recovery, while for the other two scenarios (non-edible but degradable microcarriers and edible microcarriers embedded in the final product), the edible or biodegradable materials to be used and production technologies are the primary issues [89]. In the case of scaffolds or cell-laden constructs in the maturation process, the main challenges are related to the correct mechanical stimulation cells requiring correct alignment and, eventually, mechanical tension, and an increase in material transport for efficient medium utilization, introducing recycling techniques [24,87,88]. According to Martin et al., the transition from a laboratory batch to an industrial bioreactor will require the transition from flexible bioreactors to highly specialized systems, optimized and standardized from the bioprocess point of view [24].

The industrial scaling up also represents an essential step to obtain a competitive product on the market. The first example of a cultivated burger was presented in 2013 in the Netherlands, and it required a total fabrication cost of 300.000 $ [42]. After that presentation, several companies and research groups sought to address this complex challenge. According to Guan et al., the current (2020) estimated cost for cultivated meat or fish products ranges from 66.4 $/kg to 2200.5 $/kg compared to a few dollars per kilogram for the conventional meat, and most of the cost is attributed to the cell and tissue culturing [122].

From the abovementioned discussion, the main goal of achieving cultured meat could be accomplished only if new approaches for affordable, scalable, and sustainable culturing systems will be found. These must be implemented by exploiting several design methods, including in silico models for the bioreactor production process [92]. The process must be cost-effective, hence materials used must come from abundant and animal-free sources, and the production process must be scalable, economical, and sustainable, with minimal waste production [27].

### 5. Consumer Acceptance

Although the main purpose of this review was to consider and investigate the technological and biotechnological challenges, it is necessary to emphasize that consumer acceptance plays a key role in the spread of cultured meat. Bryant at al. conducted a systematic review of several surveys on this topic [123]. This work highlighted the complexity in formulating a complete picture in people's perception of cultured meat. The different surveys reported different results. The average acceptance rate of cultured meat reported by Wilks and Philip was 63,5%, while the same parameter, identified by Hocquette, varied between 5% and 11% [124,125]. These results are discordant due to the population and sample considered, as well as the structure of the questions (how the questionnaire is formulated, e.g., willingness to try cultured meat vs. willingness to eat regularly) [126]. The most common objections relate to the unnaturalness of the product, a personal perception that is less safe and healthy than conventional meat, and an anticipated impression that the product has an inferior taste, texture, and appearance, accompanied by a higher price. In contrast, the positive arguments are related to animal welfare and environmental benefits, however, accompanied at the same time by doubts feasibility and ethical status [125]. As

suggested by Mark Post, a leading author in the field of cultured meat, customer acceptance of this product will remain speculative until this product is actually on the market [66].

## 6. Conclusions

The constant and rapid increase in the global population has led to the research to find novel protein sources to meet the increasing demand. In this scenario, cellular agriculture, specifically cultured meat, is arousing increasing interest. Cultured meat has opened an intense debate between those who see it as an innovative, ethical, and sustainable product and those who are skeptical. The full realization of this product will face multiple challenges, both from a biotechnological and technological point of view.

In the former case, the choice of animal and method for cell harvesting represent a crucial step in the large-scale production of cultured meat accompanied by the identification of FBS substitutes capable of sustaining cell viability and proliferation in both the short and long term. Although, as reported before, first steps have been taken in this direction, a completely animal-free alternative that can match the performance characteristics of FBS is still a long way from being identified.

The biotechnological approach will also be essential to create a product that is not only safe, but also reflects traditional meat. Although it is true that the changes undergone by the livestock sector have had an impact on the environment, it is equally true that these have made it possible to bring to our tables products characterized by high nutritional and functional quality. This second aspect, in addition to playing an important role in terms of consumer acceptance, represents one of the most difficult challenges to overcome. It is therefore necessary that all those organoleptic and functional characteristics, which in traditional meat are a direct consequence of animal feeding and wellbeing and are reproduced in cultured products.

From a technical point of view, the challenge is related to the implementation of a reliable and scalable process chain. The overall challenges are both related to the production and culturing systems. Concerning the production, in literature several approaches are shown to be very promising: from scaffolding, which is an older but well-known technology, to its alternatives, finally through to biofabrication and 3D bioprinting. This latter could potentially represent a game changer, even if several specific challenges must be overcome, such as, for example, the correct choice of materials as a balance of chemical, mechanical, and biological features optimized for both processability and cell cycle process compatibility. Moreover, 3DBP can lead to a closed-system and process, designed to reduce the contamination risk [33] in a scalable and modular way. Another big deal is related to the differentiation of cells within constructs trough bioreactors, which must be accomplished for the elevated size request that is different from the laboratory scale typically adopted.

In conclusion, several biotechnological and technical challenges need to be further investigated to meet quality, safety, and consumer acceptance goals. In this scenario, it is of paramount importance to promote research initiatives with an open access character to disseminate research studies, results, and solutions between public and private partners involved in the production of cultured meat.

**Funding:** This research received no external funding.

**Institutional Review Board Statement:** Not applicable.

**Informed Consent Statement:** Not applicable.

**Acknowledgments:** The authors acknowledge support from the University of Milano through the APC initiative.

**Conflicts of Interest:** The authors declare no conflict of interest.

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
