# Peer review of "Biotechnological and Technical Challenges Related to Cultured Meat Production"

_applsci, doi:10.3390/app12136771_

Round 1

Reviewer 1 Report

Manuscript professionally written and organized

Some points need to be covered:

·       Is there is a prototype (on at least Lab scale) of this cultured meat? Please clear in detail

·       Are the consumers willing to accept such cultured meat??

·       What kind of food application such cultured meat projected to use in?

·       Another crucial point needs to be included, regardless the biotechnological and technological problems, what is the economic efficiency of producing 1kg of cultured meat comparing to the conventional one?

Author Response

Dear auditor, first of all i wanted to thank you for the valuable advice and insights you have given us.

We have, inside the review, deepened all the points you highlighted.

1) Is there is a prototype of this cultured meat?

We delved into subject of the protoype and the cost of production. You can find it from line 946-952

2) Are the consumers willing to accept such cultured meat?

The subject of consumer acceptance, i think is a very important topic. We did not initially consider it because the focus of the review was on biotechnological and technical challenges. However, after your advice, we thought we introduced it and explore it further. You can find it from line 85-103.

3) Economic efficiency of producing 1 kg of cultured meat?

Please see the point1)

I hope that the points addressed in this way can be considered satisfactory.

Thank you again

Reviewer 2 Report

General comments

This paper approaches a relevant issue in an original way, bringing an important contribution. An English review, preferably by a native speaker, is needed. Text fluidity as well as structure in terms of paragraph organization must improve.

Specific Comments

Abstract: Please insert some concrete results observed in the abstract, especially an abridged list of the main challenges and the most important conclusions. Perhaps reduce the amount of introductory information in the abstract, if needed.

L17-18: Consider replacing the term health by more direct (in the case of environment) and broader terminology, specially considering human and animals - either replace by welfare or include “health and welfare”.

L31: Replace … by 2050, where it has been estimated that… by … by 2050, when it has been estimated that…, the same in L33.

L44: Replace …breeding methods… by systems if your intention is to approach all phases with a more general statement.

L47-48: Avoid repeating the same word (crisis).

L55: Reconsider the idea of classifying environmental problems as the main one: is it needed? Is it useful? Can it be used to downgrade other extremely pressing issues?

L60-62: It is essential to include the number of years described as short and long term in the paper by Lynch, as they are not the ordinary timeframe for such terms. If numbers are not mention, the context becomes distorted.

L79-80: A single-phrase paragraph is rarely adequate, reorganize the sequence of information, put all phrases which relate to the same topic in the same paragraph (Other paragraphs are extremely long - see L295-321). Please note that there are more synonyms, if this discussion is relevant to the text please refer to Table 1 at https://doi.org/10.1016/j.techsoc.2020.101286

L87-88: Why are organoleptic properties most important? Either change or add the reasons for this view.

L94: Please choose the best terminology in your view and be consistent throughout the text. Note you have used cultivated meat in the title of the paper, and in item 2 you use in vitro meat…

L118: Perhaps include energy as an important resource which is in high demand for cell-based meat as well…

L133: Use …biodegradable or edible or both… avoid and/or.

L148: …amongst different cell populations…, not ...between…

L162: Phrase seems grammatically truncated, please review.

L178-180: Note the repetitive use of the word  however… There are other opportunities for text improvement in this sense.

L185, L192: It is critical to differentiate between animal and non-animal materials throughout the whole discussion of cell-based meat production. Here, when gelatine and collagen as well as proteins in general are mentioned, a statement is required. Minimally, a warning should be inserted about the inadequacy of using animal proteins.

L195-198: Phrase is confusing, please improve.

L212: Table 1: Perhaps to the choice of the animal for cell harvesting, it could be added the challenges regarding the best cell to collect from the animal as well as the best collection method. This table is really interesting, as the organization of the challenges in a clear and straightforward way is a great contribution.

L218: Consider reserving site of collection to present as a discussion on its own - separate

L233-236 into a new item, and provide a deeper discussion.

L230-231: Please specify “… a lower concentration...” of?

L247: Do not start a phrase using acronyms.

L257: Do you intend to mention “male and female beef cattle…”? Please improve the phrase. Consider that also pregnant dairy cows may go to slaughter.

L299-305: Please approach maturation as another biotechnological challenge, and not as a definitive impossibility. This is lacking in the existing literature, as for instance the papers cited in this discussion. However, it seems that to study maturation processes for cell-based meat is not more challenging than other biotechnological breakthroughs in the production of cell-based meat. Thus, there doesn’t seem to be a good reason to consider maturation a complete impossibility in the future.

L322: Do you mean intensive animal production? Why is there a focus on breeding?

L376: Table 2: All potential medium ingredients from animal origin must be discussed in terms of their ethical and sustainability limitations. This includes sericin, royal jelly and dairy products, as well as formulations that require any percentage of FBS. It is difficult to accept that other animal-derived products may be considered alternatives to FBS, as they carry on most of the same problems. Please make this section much shorter by presenting only real alternatives, i.e., those of non-animal origin.

L499: Again the claim that sustainability is the mains challenge to win is fragile. Either remove it or expand in the introduction to explain why this is so.

L501: Cultivated meat must be a completely animal-free product, not almost completely.

L506: Do you have references to support this claim? There have been so many health crisis due to animal consumption, both food-borne diseases and new zoonotic agents, some of which you have mentioned in the introduction, as well as antibiotic resistance development due to animal production, all of which affect human health, that this statement should be put into context, if not completely rewritten.

L627-629: Why is plant tissue decellularization not included? This is a very important strategy and should be discussed.

L743: The approach to animal-based resources may be improved, to a non-tolerance stance, except for the cells of course.

L908-933: Reduce the length of this paragraph by ensuring less repetition of content already presented in previous sections.

L971: Avoid repeating introductory issues, focus on the response to the objective. The conclusion can be shorter and more straight-forward.

Author Response

Dear reviewer, first of all, i would like to thank you for your valuable comments.

We sent the article for review by a native speaker, as you recommended.

1)  Abstract

We have modified the abstract by inserting concrete results, as you suggested.

2) L30-L31

We modified "by 2050, where .." with when, as you suggested.

3) L44

We have replaced breeding methods with systems.

4)L47-L48

We have deleted the repetition of crises.

5)L55

We have developed the consept following your tips.

6)L60-L62

In the text can now be found on line 59. We have added the years as you suggested (within 20 years) and (beyond 100 years)

7)L87-L88

We have deleted the part relating to organoleptic properties.

8)L94

You can see the change now on line 111. We have standardised the term in cultivated meat.

9)L118

You can find the change now on line 134. We have addedd energy.

10)L133

We have modified the part following your tips, you can see it on line 149.

11)L148

We have replaced between with amongst, you can see now on line L164.

12)L185-L192

We have modified following your tips. You can find it on line 209-210.

13) Table 1

We have added the part on the harvesting method. We have contextualised it within the text and not put it in the table. You can find this part from line 259 to 273.

14) L230-L231

We have added "of activity of satellite cells", you can find it on line 247.

15) L247

Now L279, we have replaced FBS with Fetal Bovine Serum.

16) L257

Now L288 we have modified following your tips. "When a pregnant cow is slaughtered the fetus is removed".

17)L299-L305

Now 334-336 we have modified the part emphasising the possibility of the process.

18)L376, Table 2

 We totally agree with what you suggested, which is why we have modifed the table by removing milk, colostrum and royal jelly. However, whey protein, and sericin, we have kept them as, as waste products, they can be enhanced. However, following your advice, we have emphasised this aspect, which was missing in the previous version.

19)L499

We have removed the part following your tips.

20)L501

Now L480, We have replaced almost.

21)L506

Now L483, We have better contextualised this part, modifying inconsistencies.

22)L627-L629

Now L608, We have added plant decellularization.

22)L743

Now 744-746, we have modified as you suggested.

23)L908-933

We have modified this paragraph following your advice. We have also added a part for reviewer 1.

24) Conclusion

We have modified this part following your tips, enphasising the concrete aspects of this review

Round 2

Reviewer 2 Report

Relevant issues raised in previous suggestions to the manuscript were not addressed. Especially important is a complete English review, including overall text structure, and the superficial discussion on animal-free medium ingredients, to which the authors add animal proteins such as sericin and casein as alternatives, without an adequate approach to the fact that these are animal-based products.

Author Response

Dear Reviewer,

thank you very much for your comments and advice.

1)The text uploaded last time had already been revised by the MDPI service, you can find the certification attached. In, addition we contacted the english reviewer again and he informed us that MDPI will make a further revision if the article is published.

2)Regarding the stucture, we have shortened the paragraphs, as you recommended, and implemented a reorganisation of the text.

3)As far as alternatives to FBS are concerned, we have modified the table. We have listed several matrices, whose positive impact on cell culture make them potential alternatives, only after futher studies. The table shows plant-based, sustainable and ethical alternatives. We have retained sericin and whey protein, commented differently from the previous version.

Sericin: Being an animal derived product, its use would not fully satisfy the requirements of ethics and edibility. However, it must be reiterated that this is a waste product of the silk-processing industry and its use within a different sector can meet the circular economy principle.

Whey protein: However, it is a waste product with a high environmental pollution load...For this reason, althought it is of animal origin, its use as an alternative to FBS might be considered. 
